# Variants in Human ATP Synthase Mitochondrial Genes: Biochemical Dysfunctions, Associated Diseases, and Therapies

**DOI:** 10.3390/ijms25042239

**Published:** 2024-02-13

**Authors:** Valentina Del Dotto, Francesco Musiani, Alessandra Baracca, Giancarlo Solaini

**Affiliations:** 1Laboratory of Biochemistry and Mitochondrial Pathophysiology, Department of Biomedical and Neuromotor Sciences, University of Bologna, 40126 Bologna, Italy; valentina.deldotto2@unibo.it (V.D.D.); giancarlo.solaini@unibo.it (G.S.); 2Laboratory of Bioinorganic Chemistry, Department of Pharmacy and Biotechnology (FABIT), University of Bologna, 40127 Bologna, Italy; francesco.musiani@unibo.it

**Keywords:** mitochondria, ATP synthase, ATP6, ATP8, mt-DNA, mutations, therapy, F_1_F_o_-ATPase modeling

## Abstract

Mitochondrial ATP synthase (Complex V) catalyzes the last step of oxidative phosphorylation and provides most of the energy (ATP) required by human cells. The mitochondrial genes *MT-ATP6* and *MT-ATP8* encode two subunits of the multi-subunit Complex V. Since the discovery of the first *MT-ATP6* variant in the year 1990 as the cause of Neuropathy, Ataxia, and Retinitis Pigmentosa (NARP) syndrome, a large and continuously increasing number of inborn variants in the *MT-ATP6* and *MT-ATP8* genes have been identified as pathogenic. Variants in these genes correlate with various clinical phenotypes, which include several neurodegenerative and multisystemic disorders. In the present review, we report the pathogenic variants in mitochondrial ATP synthase genes and highlight the molecular mechanisms underlying ATP synthase deficiency that promote biochemical dysfunctions. We discuss the possible structural changes induced by the most common variants found in patients by considering the recent cryo-electron microscopy structure of human ATP synthase. Finally, we provide the state-of-the-art of all therapeutic proposals reported in the literature, including drug interventions targeting mitochondrial dysfunctions, allotopic gene expression- and nuclease-based strategies, and discuss their potential translation into clinical trials.

## 1. Introduction

Mitochondria in eukaryotic organisms host different metabolic pathways and play several roles essential to cellular life, including respiration and ATP synthesis by oxidative phosphorylation (OXPHOS), the Krebs cycle, fatty acid oxidation, and control of redox state and ROS level. However, they can also determine whether a cell should die by apoptosis [1]. These functions are almost exclusively carried out in the matrix or compartments of the inner membrane. Interestingly, mitochondria harbor their own circular genome (mt-DNA) containing 37 genes, 13 of which encode 11 polypeptides belonging to the respiratory Complexes I (CI), III (CIII), and IV (CIV), and two others belonging to the ATP synthase, or Complex V (CV): ATP6 (a subunit) and ATP8 (A6L subunit) (Figure 1A) [2,3]. Therefore, all multi-subunit OXPHOS Complexes, except Complex II (CII), require the nuclear and mitochondrial genomes to encode their corresponding subunits [3]. Exposure to high levels of reactive oxygen species (ROS) and the relatively scarce presence of mitochondrial DNA repair systems make the mitochondrial genome more prone to an increased number of somatic variants when compared to most genes of the nuclear genome in eukaryotic cells [4]. Therefore, pathogenic variants in mt-DNA increase with aging [5], and many neuronal, cardiac, neoplastic, and metabolic-related diseases are caused or at least exacerbated by mitochondrial impairment [4,6,7,8].

Mitochondrial syndromes are a heterogeneous group of rare diseases caused by genetic defects, localized in both the nuclear and the mitochondrial genomes, that cause the dysfunction of the OXPHOS system [9]. Severe neuromuscular disorders are associated with defects in ATP synthase. While pathogenic variants in nuclear genes encoding the CV subunits are very rare, those in mitochondrial genes, which often manifest in early childhood, are more frequent and better characterized, highlighting their clinical relevance [10].

Here, we describe and comment on the state of the art of the main inborn pathogenic variants in the mt-DNA genes encoding the two subunits of CV, located in the membrane domain of ATP synthase, F_o_: ATP6 and ATP8. ATP6 plays a critical role in the coupling mechanism of proton translocation to the synthesis of ATP through the rotary catalysis of ATP synthase and ATP8 on the quaternary structure of the enzyme complex. ATP6 is a hydrophobic polypeptide made of 226 amino acids in humans, which is embedded in the inner mitochondrial membrane (IMM), while ATP8 is a small hydrophilic polypeptide consisting of 68 amino acids that connects the membrane portion of the enzyme with the F_1_ catalytic domain [11,12]. It appears to play only a structural role. These two subunits are translated from the polystronic *MT-ATP8/MT-ATP6* mRNA, which presents an overlap of 46 nucleotides. The amino acid sequence of the two proteins is shown in Figure 1B,C.

This review focuses on the molecular mechanisms underlying the biochemical dysfunctions caused by ATP synthase mt-DNA variants and, for the first time, the structural changes induced by the most common variants on a human ATP synthase model. Furthermore, we considered it relevant to collect and discuss the main proposed therapeutic approaches, as reliable therapies are currently lacking. This updated review will contribute to deepening the knowledge of the molecular basis of mitochondrial diseases caused by mt-DNA variants of ATP synthase by integrating an interesting review focused on related clinical syndromes and functional consequences on a yeast ATP synthase model [13] 

## 2. ATP Synthase Structure and Mechanism of Catalysis

ATP synthase (also referred to as F_1_F_o_-ATPase or H^+^-ATPase) catalyzes the terminal step of the OXPHOS process that consists of the phosphorylation of ADP from inorganic phosphate to ATP by exploiting the energy released by the oxidation of the reduced dinucleotides NADH and FADH_2_ [3,10]. It is a ubiquitous enzyme that supplies most of the energy needed by aerobic cells. It consists of a catalytic and a membrane domain, called F_1_ and F_o,_ respectively, and forms dimers [11]. 

The monomeric structure of the enzyme has been resolved at the atomic level in some organisms including bacteria, in which it is constituted of 8 different subunits, 5 constituting the catalytic sector F_1_ (α, β, γ, δ, ε), and 3 composing the membrane embedded sector F_o_ (a, b, c) that allows proton transport through the membrane; the total number of subunits is 27 in *Escherichia coli* [14]. The mammalian mitochondrial enzyme has a more complex composition: it contains 8 homologous subunits to those of bacteria added to 10 more different subunits (18 subunits in total) on the F_o_ sector for a total of 29 polypeptide chains in humans, including the endogenous protein, IF_1_ (Figure 2) [15,16]. Indeed, in mitochondria, ATP synthase is associated with the IF_1_ protein, which inhibits the ATP hydrolytic activity of the enzyme both during its assembly [17] and under certain conditions, such as those induced by a collapse of the membrane electrochemical potential (Δμ_H+_), also called proton motive force (*pmf*) [18,19,20,21,22]. Recently, an additional binding site of IF_1_ on the OSCP subunit of ATP synthase has been described in cancer cells. The authors suggested that the interaction of IF_1_ with this site can protect cells from apoptosis [23]. 

The basic structure of the enzyme capable of synthesizing ATP at the expense of the energy released by *pmf* in coupled membranes has the composition α_3_, β_3_, γ, δ, ε, ab_2_c_15_ in *Escherichia coli*. The stoichiometry of c subunits varies from 8 in mammalian ATP synthases to 17 in bacteria. C subunits interact with one another to form a cylinder across the membrane, and the c_n_ ring pore contains phospholipids and possibly other molecules that interact with its inner surface [24]. Thus, whatever ATP synthase complex is considered, most of the F_o_ subunits are embedded in the IMM membrane, as shown in the scheme of Figure 2, but some constitute the peripheral stalk that allows F_o_ to be structurally bound to the static moiety of F_1_ made of the α and β subunits [25]. The γδε subunits of the hydrophilic moiety F_1_ constitute the central stalk of the enzyme, are strictly bound to the ring of c subunits, and form the rotor that sits at the center of the enzyme [26]. Therefore, the ATP synthases are nano-machines capable of transducing energy from electro-chemical to mechanical to chemical as the β–γ phospho-anhydride bond in ATP. In fact, the catalytic mechanism of the enzyme has been elucidated [27,28,29]: protons flowing along their gradient concentration from the intra-cristae space (ICS) to the matrix in mitochondria release energy to the rotor of the enzyme (γδε-c_n_) that is pushed to rotate. In mitochondria, protons flowing through two hydrophilic hemichannels at the interface of ATP6 (subunit a in bacteria) and the ring of c subunits induce the rotor located in the center of the three pairs of αβ subunits of F_1_, each containing a catalytic site at their interface, to rotate counterclockwise [30,31]. Since the rotor is intrinsically asymmetric, its rotation allows modifying the affinity of each of the three catalytic sites, which, in situ at the same time, never have the same conformation, but, in turn during rotation, take on the same conformations. Therefore, each catalytic site can synthesize ATP and release it. Indeed, the synthesis proceeds through three main steps: first, the substrates ADP and Pi are bound; second, the ATP synthesis reaction occurs; and finally, the product is released [27,32]. With each 360° rotation of the enzyme’s rotor, three molecules of ATP are produced and released. Since ATP6 is directly involved in the proton flow that releases energy to the enzyme rotor to catalyze ATP synthesis, variants in this protein can cause mild to even very serious diseases [33].

## 3. Biochemical Dysfunctions Related to *MT-ATP6* and *MT-ATP8* Pathogenic Variants

The mitochondrial genome presents some specific features, including maternal inheritance and heteroplasmy. Heteroplasmy is a condition in which at least two different mitochondrial genomes are present within the same cell. Pathogenic variants in the mt-DNA are highly recessive and usually coexist with the wild-type mt-DNA molecules. Therefore, the clinical manifestation of mt-DNA variants mainly depends on both their severity and the mutational load (heteroplasmy) of the tissues [2,8].

Over the last two decades, a large number of studies using different patient’s specimens and other cellular paradigms have led to in-depth investigations of the biochemical and cellular alterations caused by *MT-ATP6* and *MT-ATP8* variants, which are summarized in Table 1. We focused our attention here on three amino acids because they are the most frequently mutated in patients and have been extensively studied in recent decades.

As shown in Table 1, the cellular dysfunctions observed for a given variant can be highly variable, and one element that contributes to this characteristic is heteroplasmy.

Consequently, as with other mt-DNA-associated diseases, a specific feature is the threshold of the percentage of mutant genome (or percentage of heteroplasmy) that must be exceeded to detect a biochemical alteration. As reported below, this defect may also depend on the variant, haplogroup, cell type, and tissue type.

**Table 1 ijms-25-02239-t001:** Biochemical and cellular parameters in patient tissues and cell models carrying *MT-ATP6* and *MT-ATP8* pathogenic variants. Abbreviation: heteroplasmy (H), oxygen consumption rate (OCR), mitochondrial membrane potential (MMP), reactive oxygen species (ROS), induced pluripotent stem cell (iPSC) and neural progenitor cells (NPCs), Normal (N), Decreased (D), Increased (I), Affected (A).

Genetic Variant/Subunit AA Change	Tissue or Cell Models H (%)	Biochemical and Cellular Parameters				
		CV ATP Synthesis	CV ATP Hydrolysis	OCR	CV Assembly/Stability	Other Mitochondrial and Cellular Readouts
**m.8382C>T** **ATP8: p.T6I**	Muscle (100%) [34]		(D)			CI activity (D)
	Fibroblasts (100%) [34]	(N)	(N)			CIV activity (D)
**m.8403T>C** **ATP8: p.I13T**	Fibroblasts (100%)	(N) [34]	(N) [34]			Depolarized plasma membrane and ROS (I) [35]; CIV activity (D) [34]
	Yeast (100%) [36]	(D)			(N)	Growth in stress conditions (D); Mitochondrial membrane potential (N)
**m.8424T>C** **ATP8: p.I20P**	Muscle (100%) [34]		(D)			CI, CII, CIII, and CIV activities (D)
	Fibroblasts (100%) [34]	(D)	(N)			CI activity and growth in galactose media (D)
	Cybrids (100%) [34]	(D)	(D)			CI and CIV activities (D); Lactate production (I)
**m.8528T>C** **ATP8: p.W55R** **ATP6: p.M1T**	Fibroblast (93%) [37]	(D)				
	Heart muscle (90%) [38]				(A)	CV subunit levels and CI activity (D); ATP6 and ATP8 protein levels (D)
**m.8529G>A** **ATP8: p.W55X** **ATP6: p.M1M**	Muscle (>90%) [39]	(D)	(D)		(A)	CI-CIV activities (N)
	Fibroblast (>90%) [39]		(D)			CI-CIV activities (N)
	Cybrids (100%)		(D) [39,40]	(D) [40]	(A) [39,40]	Growth in galactose media (D); ATP6 and ATP8 protein levels (D); Complexes II, III, and IV levels (D) [40]
**m.8561C>G** **ATP8: p.P66A** **ATP6: p.P12R**	Myoblasts (99%) [41]				(A)	Total ATP level (D); ROS (N); ATP6 and ATP8 protein levels (N)
**m.8561C>T** **ATP8: p.P66L** **ATP6: p.P12S**	Muscle (99%) [42]		(D)		(A)	
**m.8611insC** **ATP6: p.L29PfsX36**	Muscle (60%) [43]		(D)		(A)	
	Fibroblasts (80%) [43]		(D)		(A)	ATP6 protein level (D); Mitochondrial cristae structure and dynamics (A)
**m.8618insT** **ATP6: p.T33HfsX32**	Muscle (65–85%)				(A) [44,45]	ATP6 protein level (D) [44]
	Fibroblasts (45%) [45]			(D)	(A)	ROS (I); Mitochondrial network morphology (N)
**m.8648G>A** **ATP6: p.R41Q**	Fibroblast (100%) [34]	(N)	(N)			
	Cybrids (100%) [34]	(N)	(N)			
**m.8782G>A** **ATP6: p.G86X**	Fibroblasts (12–27%) [45]			(D)	(A)	ROS (I); Mitochondrial morphology (N)
**m.8806C>G** **ATP6: p.P94A**	Muscle (100%) [34]	(N)				CI-CIV activities (D)
**m.8839G>C** **ATP6: p.A105P**	Cybrids (100%) [46]	(N)				Growth in galactose media (D); Mt-DNA copy number (I); OXPHOS protein levels (I); Mitochondrial membrane potential (D); CI-CIV activities (N)
**m.8843T>C** **ATP6: p.I106T**	Yeast (100%) [47]	(N)		(N)	(N)	Mitochondrial membrane potential (N)
**m.8851T>C** **ATP6: p.W109R**	Yeast (100%) [48]	(D)	(D)	(D)	(N)	Growth in stress conditions (D); Mitochondrial cristae structure (A); CIII and CIV super-complexes (D)
**m.8909T>C** **ATP6: p.F128S**	Yeast (100%) [49]	(D)		(D)	(A)	
**m.8932C>T** **ATP6: p.P136S**	Yeast (100%) [50]	(D)		(D)	(A)	ATP6 protein level (D)
**m.8946A>C** **ATP6: p.M140I**	Fibroblasts (100%) [34]	(N)	(N)			CI activity (D)
**m.8950G>A** **ATP6: p.V142I**	Lymphocytes [51]			(D)		
	Yeast (100%) [47]	(D)		(D)	(N)	Sensitivity of growth to oligomycin (I); Mitochondrial membrane potential (N)
**m.8969G>A** **ATP6: p.S148N**	Muscle (100%) [34]	(N)				CI activity (D)
	Yeast (100%)	(**D**) [52,53]	(D) [52]	(D) [52,53]	(A) [52]	Growth in stress conditions (D) [52,53]
	Cybrids (19–98%) [52]			(D)		Mitochondrial cristae structure (A); ROS (I)
	Fibroblasts (100%) [54]			(D)		
**m.8975T>C** **ATP6: p.L150P**	Muscle [34]		(D)			CI activity (D)
	Fibroblasts (100%) [34]	(**D**)	(N)			CI activity (D); Growth in galactose media (D)
	Cybrids (100%) [34]	(**N**)	(N)			CI and CIV activities (D); Growth in galactose media (D); Lactate production (I)
**m.8989G>C** **ATP6: p.A155P**	Muscle (92%) [55]		(D)			Mitochondrial ultrastructure (N)
**m.8993T>G** **ATP6: p.L156R**	Yeast (100%) [56]	(**D**)	(D)	(D)	(A)	Growth in stress conditions (D); CIV level (D)
	Platelets (80–93%)	(**D**) [57,58]	(N) [57,58]			CV ATP-driven proton flow (N) [57]
	Lymphocytes (80–100%)	(**D**) [59,60,61,62]	(D) [59]	(D) [60]		ROS and mitochondrial membrane potential (I) [62]; CV proton flow (D) [61,63]; Oligomycin sensitivity of CV proton flow (I) [63]
	Muscle (76%) [64]				(A)	
	Fibroblasts (70–100%)	(**D**) [34,65,66,67,68,69,70]	(D) [66,69,71]; (N) [34,65,67]	(D) [72,73]; (N) [65]	(N) [67,68]	Mitochondrial membrane potential (I) [67,70]; Mitochondrial morphology (A) [70,73]; ROS (I) [70,74]; Antioxidant enzymes (A) [70]; Oligomycin sensitivity of CV (I) [65]; Growth in galactose media (D) [34,66,69,71]; Mitochondrial calcium uptake (D) [70]; Glycolytic capacity (D) [73]; CI and CIV activities (D) [34]
	Cybrids (45–100%)	(D) [34,66,68,71,75,76,77,78,79,80]	(D) [34]; (N) [71]	(D) [60,71,77,78,79,81,82]	(A) [68,75,79]	Mitochondrial membrane potential (D) [81] or (I) [78,80]; Mitochondrial morphology (A) [83,84]; Mitochondrial ultrastructure (A) [80]; ROS (I) [78,81,85]; Antioxidant enzymes (A) [78,85]; Growth in galactose media (D) [66,76,78]; ATP level (D) [81]; Extracellular lactate (I) [34,77]; Autophagy (I) [84]; CI, CII, or CIV activities (D) [34,78,79]; Oligomycin [68] and apoptosis [80] sensitivity (I); Actin cytoskeleton and Ca2+ in-flux rates (A) [83]; Reductive carboxylation of glutamine and NADH/NAD ratio (I) [82,86]
	IPSCs (90–100%)			(D) [87]; (N) [88]		Mitochondrial membrane potential, ROS, and lactate production (I) [89]
	NPCs, Neurons (90–100%)			(D) [89]		Mitochondrial membrane potential, ROS, and antioxidant enzymes (I) [89]; Degenerative defect [89]; Metabolic dysregulation; Formation of cerebral organoid (A) [88]
**m.8993T>C** **ATP6: p.L156P**	Yeast (100%) [90]	(D)	(N)	(D)	(N)	CIV level, COX2, and ATP6 protein levels (D)
	Lymphocytes (90–95%)	(D) [62]				Mitochondrial membrane potential (N); ROS (I) [62]; Proton flux (D) [63]
	Fibroblasts (95–100%)	(D) [34,65]	(D) [34], (N) [65]	(N) [65]	(N) [68]	Depolarized plasma membrane and ROS (I) [35]; Growth in galactose media (D) [34]
	Cybrids (100%)	(D) [71,77]; (N) [34,79]	(N) [34]	(D) [77]; (N) [79]	(N) [68,79]	Lactate production (I) [34,77]
**m.9008C>G** **ATP6: p.T161S**	Muscle (100%) [34]	(N)				
	Fibroblasts (100%) [34]	(N)	(N)			CI activity (D)
	Cybrids (100%) [34]	(D)	(N)			Growth in galactose media (D); Lactate production (I)
**m.9016A>G** **ATP6: p.I164V**	Yeast (100%) [47]	(N)		(N)	(N)	Mitochondrial membrane potential (N)
**m.9019A>G** **ATP6: p.T165A**	Muscle (100%) [34]	(D)				CI activity (D)
**m.9025G>A** **ATP6: p.G167S**	Yeast (100%) [47]	(D)		(D)	(N)	Sensitivity of growth to oligomycin (I); Mitochondrial membrane potential (N)
**m.9029A>G** **ATP6: p.H168R**	Yeast (100%) [47]	(D)		(D)	(N)	Sensitivity of growth to oligomycin (I); Mitochondrial membrane potential (N)
	Cybrids (100%) [81]	(D)				ATP level (D); ROS and mitochondrial membrane potential (I)
**m.9032T>C** **ATP6: p.L169P**	Cybrids (25–80%) [81]	(D)				ATP level (D); ROS and mitochondrial membrane potential (I)
**m.9035T>C** **ATP6: p.L170P**	Cybrids (100%) [91]		(D)			ROS and antioxidant enzymes (I); Mitochondrial membrane potential (N); Sensitivity to glucose deprivation (I); Oxidative stress (I)
	Muscle (100%) [34]	(D)				CI activity (D)
	Fibroblasts (100%)	(D) [34]	(N) [34]	(D) [92]	(A) [92]	Growth in galactose media (D) [34]
**m.9058A>G** **ATP6: p.T178A**	Yeast (100%) [47]	(N)		(N)	(N)	Mitochondrial membrane potential (N)
**m.9101T>C** **ATP6: p.I192T**	Lymphocytes (100%) [93,94]	(D)				
	Cybrids (100%) [94]	(D)				
**m.9127** **delAT** **ATP6: p.I201PfsX2**	Fibroblasts (50%) [95]	(D)	(D)	(N)		Oligomycin-induced increase in mitochondrial membrane potential (D)
**m.9134A>G** **ATP6: p.E203G**	Muscle [96]	(D)	(D)			
**m.9139G>A** **ATP6: p.A205T**	Yeast (100%) [47]	(N)		(N)	(N)	Mitochondrial membrane potential (N)
**m.9154C>T** **ATP6: p.Q210X**	Fibroblasts [97]			(N)	(A)	Mitochondrial morphology (A)
	IPSC and Neurons [97]				(A)	Motor neuron differentiation (A); Mitochondrial morphology (A); Hyperactivation of the Notch pathway
**m.9160T>C** **ATP6: p.Y212H**	Yeast (100%) [47]	(N)		(N)	(N)	Mitochondrial membrane potential (N)
**m.9176T>G** **ATP6: p.L217R**	Yeast (100%) [98]	(D)		(D)	(A)	Growth in stress conditions (D); CIV super-complexes (D); ATP6, COX2, and CYTB protein levels (D); Mitochondrial ultrastrucure (A)
	Muscle (>95%) [64]				(A)	
	Fibroblasts (95–100%)	(D) [99]	(N) [71]		(N) [99]	Mitochondrial membrane potential (I) [99]; Growth in galactose media (D) [71]
	Cybrids (30–100%)	(D) [71,79,80]	(N) [71]	(D) [71,79]	(A) [79]	CI and CIV activities (D) [79]; Mitochondrial ultrastructure (A) [80]; Mitochondrial membrane potential and apoptosis sensitivity (I) [80]
**m.9176T>C** **ATP6: p.L217P**	Yeast [100]	(D)	(N)	(D)	(A)	
	Muscle (100%) [34]	(D)				
	Fibroblasts (100%)	(N) [101]; (D) [102]	(N) [71]		(A) [102]	Mitochondrial network morphology (N) [102]; Depolarized plasma membrane and ROS (I) [35]
	Cybrids (100%) [71]	(D)	(N)			
**m.9185T>C** **ATP6: p.L220P**	Yeast [103]	(D)	(N)	(N)	(N)	Sensitivity of growth to oligomycin (I)
	Muscle (>97%)	(D) [104,105]			(A) [105]	CI, CII, and CIV activities (N) [106,107,108]
	Lymphocytes [106]	(N)	(D)			
	Fibroblasts (90–100%)	(D) [34,35]	(D) [35]; (N) [34]	(D) [35]	(N) [35]	CI activity (D) and depolarized plasma membrane [35]; ROS or antioxidant enzymes (I) [35,109]; CI, CII, and CIV activities (N) [106]; Mitochondrial membrane potential (N) [109]; Lactate production (I) [34]
	Cybrids (100%)	(D) [109]; (N) [34]	(D) [35]; (N) [34]	(D) [35]		CI activity (D) [35]; Lactate production (I) [34]; Mitochondrial membrane potential (N) [109]
	NPC and neuron (100%) [109]	(D)		(N)		Mitochondrial membrane potential (I); Mitochondrial calcium homeostasis (A); Depolarized plasma membrane; Mitochondrial cristae structure and ROS (N)
**m.9191T>C** **ATP6: p.L222P**	Muscle (94%) [104]	(D)		(D)		
	Yeast [103,110]	(D)	(N)	(D)	(A)	Growth in stress conditions (D); CIV level (D); ATP6 protein level (D)
**m.9205delTA** **ATP6: p.X227NA**	Muscle (>98%) [111]					CIV activity (D)
	Fibroblasts (>98%)	(D) [111]	(N) [111]	(D) [111]	(A) [111]	CIV activity (D) [111,112]; ATP6 protein and CIV subunit levels (D) [111]; Morphological abnormalities of mitochondria [112]

### 3.1. The mt-DNA Pathogenic Variants at Position m.8993

The two most common variants in *MT-ATP6* are m.8993T>G (p.Leu156Arg) and m.8993T>C (p.Leu156Pro), which cause a change in a highly conserved leucine residue on ATP6 [13,57,113]. These variants are the most common and are responsible for approximately 50% of reported *MT-ATP6* disease cases [33]. These variants are associated with Neuropathy, Ataxia, and Retinitis Pigmentosa (NARP) or Maternal Inherited Leigh syndrome (MILS) when heteroplasmy is between 70 and 90% or greater than 90%, respectively. Furthermore, the T>G transversion usually results in a more severe clinical phenotype than the T>C transition [13,113,114,115]. 

#### 3.1.1. Biochemical and Cellular Dysfunctions in Mutated Cell Models

Analyses of patient specimens carrying the m.8993T>G or the m.8993T>C variant have been performed in platelets [57,58], lymphocytes [59,60,61,62,63], muscle tissue [64], and skin fibroblasts [34,35,65,66,67,68,69,70,71,72,73,74,116] (see Table 1). Several biochemical abnormalities have been identified, including a decreased ATP synthesis and oxygen consumption rate (OCR) [34,57,58,59,60,61,62,65,66,67,68,69,70,71,72,73], often in direct correlation with the mutation load [58,61,71,76], an alteration of the proton flux [61,62,63], and a not-fully assembled CV [64]. In human cells, these ATP synthase dysfunctions lead, as secondary effects, to a reduction in growth in stress medium [34,66,69,71], to an increase in both mitochondrial membrane potential (MMP) [62,67,70,116] and ROS [35,62,70,74], as well as to an altered mitochondrial network morphology and cristae structure [70,73]. 

In cells of NARP patients carrying the m.8993T>G variant, the severe impairment of OXPHOS has been proposed as the primary pathogenic defect; instead, the increase in ROS could be the main contributor to the pathogenesis of the disease associated with the m.8993T>C variant [62]. 

It is worth noting that no significant effects of the m.8993T>G variant have been reported on either ATP hydrolytic activity or ATP-driven proton transport by Complex V in patient cells [57,67]. However, inhibition of ATP hydrolytic activity could contribute to energy preservation and survival of cells under stress conditions (oxygen shortage), leading to the collapse of the proton motive force and ATP synthase working in reverse. Incidentally, due to the heterogeneity of the membrane potential within the same mitochondrion and the possible coexistence of ATP synthase working physiologically and in reverse [117,118], patients might benefit from the use of a specific inhibitor of the hydrolytic activity of CV.

The different percentages of heteroplasmy and other factors, including nuclear background and the type of tissue analyzed, may contribute to the phenotypic differences observed in the analysis of patients’ specimens, as well as those observed in clinical outcomes [33,92,119]. For these reasons, transmitochondrial cybrids are widely used to validate the possible pathogenicity of a mitochondrial variant, even in homoplasmic populations, with the advantage of using a cell model with the same nuclear background [120]. In the case of the two *MT-ATP6* variants, this cell model clearly demonstrated the impairment of respiration [60,71,77,78,79,81,82], ATP synthesis [34,66,68,71,75,76,77,78,79,80], mitochondrial morphology [83,84], and enhanced ROS production [78,81,85], confirming the milder effect of m.8993T>C compared to the T>G variant [62,68,71,77,79].

Interestingly, analysis of different cybrid lines carrying the same m.8993T>G variant highlighted that the mitochondrial genome sequence, and thus the haplogroup, is a factor contributing to the variations in the observed biochemical phenotypes, ranging from normal to severe defects [79]. Moreover, clear evidence of the role of mutation load on deleterious biochemical abnormalities has been recently reported in isogenic cybrids, where the OCR reduction and the extracellular acidification rate (ECAR) increase were proportionally linked to the levels of heteroplasmy, indicating a switch toward glycolysis [82]. Metabolic remodeling induced by the m.8993T>G variant was also investigated, and both proteomics and metabolomics analysis were consistent with increased glycolysis and reductive carboxylation of glutamine to support cell survival and to maintain redox balance [82]. Accordingly, a second report showed that, in cybrids, the impaired OXPHOS activity induces compensatory energy-generating anaplerotic mechanisms where glutamine-glutamate-α-ketoglutarate metabolism sustains cell survival [86]. 

The deleterious mechanism hypothesized based on all these studies, especially for the m.8993T>G variant, includes defective proton transport across F_o_, failure of the enzyme to couple phosphorylation of ADP on F_1_ to proton flow, or alteration of the holoenzyme assembly and stability [33,61,62,63]. Considering that these alterations have been observed in different cellular models despite not always being together, it seems reasonable that all three mechanisms contribute to the pathogenicity of the variants. 

In recent years, the introduction of induced pluripotent stem cell (iPSC) technology allowed disease modeling by overcoming the difficulty of accessing clinically relevant patients’ cells or tissues, such as neurons. The generation of IPSCs requires multiple quality checks and presents the issue of heteroplasmy fluctuations due to the genetic bottleneck occurring in the reprogramming process. In addition, the mutant load can also change during differentiation or cell culture and therefore must be constantly monitored [121,122]. 

A series of patient-derived iPSCs carrying the m.8993T>G or T>C variant has been developed [89,123,124,125,126], and neural progenitor cells (NPCs) and neurons have been differentiated [89]. The mutant iPSCs were able to differentiate into the three embryonic germ layers (endoderm, mesoderm, and ectoderm) [109,125,126]. However, analysis of embryoid bodies showed impaired differentiation potential in cells with a high percentage of the variant [125]. Overall, the generated cell types recapitulate the energy defects observed in other cell models and the degenerative phenotypes observed in patients [87,89]. Neurons, in part because of their predominantly mitochondria-dependent oxidative metabolism [89,109], have shown degenerative defects not detectable in other, less differentiated cells, notably ATP shortage and AMPK activation, finer neuronal fibers, and increased sensitivity to glutamate toxicity [89]. Furthermore, a study of mutant IPSCs revealed abnormalities during the three-dimensional differentiation and a defective formation of cerebral organoids, particularly in the generation of neural epithelial buds, as well as impaired corticogenesis with an altered metabolic profile [88].

#### 3.1.2. Modeling of ATP6 Subunit Carrying Changes in Leu156 in the ATP Synthase Human Structure 

In the recently released ATP synthase human structures ([16], PDB id 8H9S, 8H9T, 8H9U, and 8H9V for states 1, 2, 3a, and 3b, respectively), the Leu156 residue is located on helix H5 and is buried in the core of ATP6, forming van der Waals contacts with Leu217 and Val218, located on helix H6 (Figure 3). In the four structures, the residues in the vicinity of Leu156 are the same, and there are no conformational transitions involving ATP6 in the different states of the human ATP synthase. The variant of Leu156 in arginine can damage the interaction between subunit helices H5 and H6 because of (i) the larger volume of an arginine residue with respect to a leucine and (ii) the presence of a charged side chain in a region populated only by hydrophobic residues. In other words, the presence of an arginine in position 156 can cause a divarication between helices H5 and H6 that in turn can alter the folding of ATP6. Indeed, the correct positioning of these helices is crucial for proton access to the negatively charged Glu58 in the c subunits (Figure 3). As a consequence, we hypothesize that this may affect the proton flow and its coupling to the synthesis of ATP. The p.Leu156Pro variant can also cause some sort of damage to the fold of helix H5, but in a region that is less important for proton translocation. Indeed, the proline is a well-known helix terminator and a variant of Leu156 to proline can cause a steric clash between Pro156 Cδ carbon and the backbone oxygen atom of Gln152 and Pro153 together with the loss of a hydrogen bond between Leu156 and Gln152 backbone. On the other hand, proline is a hydrophobic residue smaller than leucine. Then, the Leu156Pro variant should not cause large damages to the hydrophobic core or to the global fold of ATP6.

### 3.2. The Pathogenic Variants at Nucleotide m.9176

The m.9176T>C and m.9176T>G variants, which cause a Leu217Pro and a Leu217Arg ammino acid change, are frequently found in MILS patients and were first reported in 1995 and 2001, respectively [99,101]. 

In the case of the m.9176T>G variant, a partially disassembled CV, decreased ATP synthesis, and mitochondrial respiration due to a defective OXPHOS pathway led to an increase in MMP in MILS patient-derived fibroblasts [71,99]. Similar observations have been described in cybrids [71,79,80] and in patient-derived muscle tissue [64]. Furthermore, analysis of this variant in yeast highlighted a severe reduction in the ATP6 protein level, suggesting that it may affect the assembly of the ATP synthase complex and cause mitochondrial and bioenergetic dysfunctions [98]. Human IPSCs have been recently generated for the m.9176T>G [126], and their use to develop neurons will be instrumental in deeply characterizing the pathogenic mechanism of this variant in a disease-target tissue. 

In the first report in 1995, biochemical analysis of the m.9176T>C variant revealed no defects in ATP synthase function in patient cells with the homoplasmic variant [101]. However, further studies reported impaired ATP synthesis [71,100,102] and CV stability [102] in both human cells and mutant yeast. As for the variants at the nucleotide m.8993, the alteration caused by the T>C variant was less severe than that caused by the T>G variant [71,100]. 

In human structures, Leu217 is positioned in helix H6 of ATP6 and is part of both the interface between helices H5 and H6 and between ATP6 and the c_8_-ring. While the residues in ATP6 in the vicinity of Leu217 (Leu156, Arg159, Val213, Leu216, Leu220, and Tyr221) do not change during the catalytic cycle, the residues found close to Leu217 in the c subunit are different depending on the ATP synthase state. Indeed, in state 1, Leu52 and Leu56 in the c subunit are close to Leu217 in ATP6 (Figure 4A), while in states 2 and 3a, Leu217 is in the vicinity of Leu52 and Ala55 in the c subunit (Figure 4B). Finally, in state 3b, no residue from any c subunit is in the closeness of Leu217 (Figure 4C). The variant of Leu217 in an arginine residue appears to cause two effects: (i) arginine is a larger residue with respect to leucine, causing some sort of friction between the ATP6 and the c_8_-ring that is rotating during the catalytic cycle, and (ii) the arginine has a charged side chain that can form H-bonds with other residues in the vicinity, such as Tyr221 from ATP6. This newly formed H-bond can interfere with the formation of another H-bond between Tyr221 and a water molecule in the outlet proton translocation half-channel. The latter water molecule is held in the correct position by two H-bonds, the already cited one with Tyr221 and a second with Glu58 from the c subunit. On the other hand, the variant of Leu217 in a proline residue can cause some sort of effects on the folding of helix H6 downstream of the mutated residue, but—as in the case of p.Leu156Pro—the damage caused by the presence of a small hydrophobic residue in position 156 should be moderate.

### 3.3. The mt-DNA Pathogenic Variant at Position m.9185

The variant at nucleotide m.9185T>C (p.Leu220Pro) was first reported in 2005 [104], and functional studies revealed a moderate effect of this variant on ATP synthase functioning. Indeed, in patient cells, besides normal Complexes I-IV activity [104,107,108], a slight alteration of CV and a depolarization of the plasma membrane were reported, often only in the case of homoplasmy [34,35,104,105,106,109]. Mild effects on ATPase function have also been observed in mutated cybrids and yeast cells [34,35,103,109]. The evaluation of IPSCs and their patient-derived NPCs [109,127], in addition to the defective ATP production, allowed us to highlight mitochondrial impairment that is hidden in the other cell types. Indeed, neural cells presented a mitochondrial hyperpolarization and an alteration of mitochondrial calcium homeostasis, as evidenced by both transcriptomic and proteomic analysis [109]. All these data suggest that the variant may alter the ability of these cells to produce ATP and control MMP, causing neural impairment. 

In the human ATP synthase structures, Leu220 is located on helix H6 of ATP6, just one helix turn away from Leu217, and forms van der Waals contacts with Met60, located on helix H3. Except for Glu224, the other residues found in the vicinity of Leu220 in ATP6 are all hydrophobic (Leu216, Leu217, and Tyr221). As for Leu217, Leu220 is at the interface between ATP6 and the c_8_-ring, and the interacting residues from the latter depend on the ATP synthase state. Leu220 of ATP6 is in the vicinity of some residues of the c subunit: Phe47 and Ile51 in states 1 and 3b (Figure 5A) and Leu52 in states 2 and 3a (Figure 5B). As for the previously discussed cases, a variant of Leu220 in a proline residue can have some effects on the fold of helix H6 downstream of the mutated residue and, in turn, cause some problems to the ATP synthase mechanism. On the other hand, proline is a small hydrophobic residue, and no serious steric or electrostatic effects are expected.

### 3.4. Other MT-ATP6 and MT-ATP8 Pathogenic Variants

A number of *MT-ATP6* and *MT-ATP8* variants reported in the literature have been reviewed by two different research groups [13,33], and the functional studies of cell models are reported in detail in Table 1. Here, we report and describe the recently identified variants. 

The study of the m.8909T>C variant, found in a patient also carrying the pathogenic m.3243A>G variant in mt-tRNALeu (MT-TL1), reported a defect in Complex V assembly and ATP synthesis [49]. A compromised assembly of ATP synthase and a reduced OCR has been observed in fibroblasts of two patients carrying the same m.8782G>A variant, one presenting adult-onset cerebellar ataxia, chronic kidney disease, and diabetes, whereas the other had myoclonic epilepsy and cerebellar ataxia [45].

The truncating variant m.9154C>T was found in a patient with adult-onset axonal neuropathy, ataxia, and IgA nephropathy and caused alteration of Complex V assembly, mitochondrial morphology, and ultrastructure in mutated fibroblasts [97]. Interestingly, the mutation load resulted to be proportional to Complex V assembly defect in patient-derived iPSCs and responsible for impaired neurogenesis due to Notch hyperactivation and altered metabolism of mature motor neurons [97].

Other identified *MT-ATP6* variants include the m.8858G>A variant in a sporadic case of NARP-MILS [128]; the m.8936T>A in a young boy with atypical mitochondrial Leigh syndrome associated with bilateral basal ganglia calcifications [129]; m.9143T>C in a patient with insulin-dependent diabetes mellitus, recurrent lactic acidosis, infections, and immunodeficiency [130]; m.9154C>T in a patient with neuropathy, cerebellar ataxia, and IgA nephropathy [131]; and m.9171A>G in a patient with mitochondrial retinopathy with atrophy [132]. The three variants m.8572G>A, the m.8578C>T and m.8812A>G were found in patients with adult-onset spinocerebellar ataxia (SCA) [133].

Regarding new variants affecting *MT-ATP6*, *MT-ATP8,* or both, m.8561C>T, which causes a defect in CV assembly, was reported in a child with early onset ataxia, psychomotor delay, and microcephaly [42], whereas functional studies have been performed for the m.8382C>T, m.8424T>C, m.8806C >G, m.8975T>C, m.9008C>G, and m.9019A>G variants [34].

## 4. *MT-ATP6* and *MT-ATP8* Variants and Clinical Phenotypes

The first variant in the mt-DNA affecting ATP6 of ATP synthase was described in 1990 as m.8993T>G on *MT-ATP6*, and found in four members of a family affected by NARP syndrome [134]. This variant is the most common variant associated with NARP, an adult-onset slowly progressive disease, whose non-canonical clinical manifestations include cerebral or cerebellar atrophy, optic atrophy, cognitive and hearing impairment, dementia, renal insufficiency, epilepsy, and diabetes [135]. 

Since then, a very large and still growing number of different *MT-ATP6* and *MT-ATP8* variants have been reported in mitochondrial patients, with variants located in *MT-ATP8* being less frequent. Patients present a highly variable clinical phenotype, ranging from asymptomatic to multisystemic neurodegeneration, and the onset of the manifestation can occur both in the pediatric population and in adults [13,33]. 

MILS, which is a subset of Leigh syndrome, is a second common disorder associated with pathogenic *MT-ATP6* variants whose first variant was reported in 1992 [136]. MILS is a highly disabling disease with an early onset (predominantly age < 2 years), characterized by bilateral lesions in the central nervous system, in association with developmental delay and regression, movement disorders, hypotonia, ataxia, dystonia, ophthalmologic abnormalities, and other multisystemic symptoms [9,137]. Differently from NARP, MILS is a very severe disorder with a poor outcome, with 50% of affected individuals dying within 3 years of age [138,139].

In the last three decades, a high variability of clinical manifestations has been observed in patients carrying *MT-ATP6* variants, including bilateral striatal necrosis [101,140], Leber hereditary optic neuropathy (LHON) [93], adult-onset ataxia and polyneuropathy [141], schizophrenia [142], hereditary spastic paraplegia [102], spinocerebellar ataxia [108], Charcot–Marie–Tooth disease [105], mitochondrial myopathy, lactic acidosis and sideroblastic anemia (MLASA) [54], and motor neuron syndrome [143]. The less common variants in *MT-ATP6/8* or *MT-ATP8* reported in the literature showed to cause maternally inherited diabetes and deafness syndrome [144], left ventricular hypertrabeculation [145], brain pseudoatrophy and mental regression [146], cardiomyopathy [39], autism [147], epilepsy [148], ataxia, peripheral neuropathy, diabetes mellitus, and hypergonadotropic hypogonadism [41]. Recently, early onset ataxia, psychomotor delay and microcephaly [42], leukodystrophy, renal disease, and myoclonic epilepsy with cerebellar ataxia [45] have been added to the already wide range of clinical symptoms. Furthermore, the clinical manifestations are often highly variable. For example, *MT-ATP6*-related Leigh disease has been reported in cases with adult onset [149,150] or in patients with MRI findings that differ from classic MILS, presenting delayed myelination, cerebral atrophy/microcephaly, or no pathology [151,152]. In addition, new biochemical dysfunctions and disease symptoms have been added to the canonical phenotypic spectrum related to *MT-ATP6* variants, such as carboxylase deficiency [153] or recurrent infections and immunodeficiency [130]. 

One of the most important factors responsible for the high heterogeneity of the disease phenotype is heteroplasmy. As with the biochemical phenotype, a specific threshold of mutational load must be exceeded to see clinical manifestations. The value of the threshold may depend on the nature of the variant, the cell type, and is usually greater than 50% [9,154]. The type of the symptoms often depends on the degree of heteroplasmy. For example, the same *MT-ATP6* variant can cause MILS or NARP syndromes if the heteroplasmy found in stable tissues is higher than 90% or between 70 and 90%, respectively [9,33]. However, the degree of heteroplasmy observed in patients is not always strictly correlated with the severity of the clinical phenotype since high percentages of mutations can occur even in asymptomatic individuals [33,151]. 

Moreover, different disease phenotypes have been reported in patients with the same genotype and belonging to the same family [151]. Therefore, other factors such as variable penetrance, nuclear background that may contribute by means of unknown nuclear variants, mt-DNA haplogroup, as well as age and sex, may affect the puzzling clinical variability that characterizes *MT-ATP6*- and *MT-ATP8*-related diseases.

## 5. Therapeutic Approaches

ATP synthase dysfunctions are crucial in neurodegenerative pathologies related to *MT-ATP6* and *MT-ATP8* variants and can lead to secondary biochemical defects associated with clinical phenotypes. 

Various therapeutic strategies have been developed for *MT-ATP6/8* diseases: (i) to target the metabolic dysfunctions caused by the variants, (ii) to increase the level of the wild-type ATPase subunit, or (iii) to decrease the mutation load, as shown in Figure 6.

### 5.1. Targeting Mitochondrial Dysfunctions

In homoplasmic m.8993T>G cybrids, our research group demonstrated that supplementation of both α-ketoglutarate (αKG) and aspartate contributed to significantly increased ATP levels and cellular survival. Indeed, the succinyl coenzyme A (CoA) produced by the α-ketoglutarate dehydrogenase catalyzed reaction was converted into succinate, producing ATP via the substrate-level phosphorylation mechanism [155]. Therefore, the combined supplementation of these substrates has been proposed as a dietary therapeutic approach for patients with severe ATP synthase dysfunction [155]. Interestingly, metabolomic analysis of the same mutated cybrids revealed their high dependence on glutamine-glutamate-α-ketoglutarate and confirmed that the supplementation with dimethyl-αKG, a membrane-permeable analog of α-KG, can activate anaplerotic energy metabolism and support cell growth. This study suggests the dietary strategy as a therapeutic option for other mitochondrial myopathies [86]. 

In many pathological conditions, ROS generation is often increased due to dysfunction in the OXPHOS system, leading to irreversible cellular injury; thus, redox-active molecules are widely used in the treatment of mitochondrial diseases [156,157]. The enhanced ROS production observed in almost all the analyzed *MT-ATP6/MT-ATP8* mutated cell types suggested the use of antioxidant molecules to prevent ROS-induced damage [62,78]. The efficacy of N-acetylcysteine (NAC), a molecule that supplies the intracellular pool of reduced glutathione, was tested in vitro in both m.8993T>G cybrids and fibroblasts, where it led to a reduction in ROS and an improvement in mitochondrial respiration and ATP synthesis [78]. In NARP patient-derived fibroblasts, the vitamin E-derivative trolox diminished both mitochondrial and cytosolic superoxide anions, but no biochemical studies on OXPHOS dysfunctions have been performed [70]. Selenium supplementation showed a mild effect in cybrids. This molecule induced an increase in the NRF1 protein level, a transcription factor that regulates the expression of antioxidant enzymes, in glutathione peroxidase and thioredoxin reductase activity and a consequent drop in the ROS level, with no changes in the amount of respiratory chain proteins [85,158]. Melatonin, a molecule that can act as an endogenous free radical scavenger and plays an effective role in preserving functions, has been shown to protect m.8993T>G cybrids from stress-induced cell death, cardiolipin depletion, and alterations in mitochondrial movement [159]. Finally, a low dose of resveratrol (10 nM), a compound that promotes mitochondrial biogenesis and exhibits antioxidant activity, was reported to boost mitochondrial respiration in a homoplasmic m.8993T>G patient’s fibroblasts [72]. 

EPI-743, also known as vatiquinone, is a para-benzoquinone analog formed by the combination of coenzyme Q_10_ (CoQ_10_) and vitamin E and presents improved both pharmacological properties and antioxidant efficacy compared to CoQ_10_ or idebenone [160,161]. The antioxidant activity is based on its ability to undergo a reversible two-electron cycling reaction, which leads to an increase in the concentrations of reduced glutathione (GSH) and an improvement in the cellular redox status [160,161]. EPI-743 has been used in several trials for heterogeneous groups of neurological disorders [160,162]. Interestingly, this drug has been used in children with genetically confirmed Leigh syndrome in an open-label phase 2A trial, showing improvement in clinical outcomes [163], as well as in randomized, double-blind, placebo-controlled trials (ClinicalTrials.gov ID: NCT01721733 and NCT02352896) [162]. Furthermore, a phase 3 clinical trial to test vatiquinone on patients with inherited mitochondrial disorders, including Leigh syndrome, is active and enrolling patients (ClinicalTrials.gov ID: NCT05218655).

The strategy of reducing ATP deficit by limiting the energy-consuming processes has been used to treat several mitochondrial diseases [156]. The Mammalian Target of Rapamycin Complex 1 (mTORC1) plays a central role in the regulation of various processes. In response to environmental conditions, mTORC1 stimulates anabolism and mitochondrial energy production and inhibits catabolic processes such as autophagy [164,165]. Although no changes in the mTORC1 pathway was observed in m.8993T>G cybrids [166], the MILS patient-derived neurons exhibited enhanced mTORC1 activity, as shown by increased phosphorylation of its targets [89]. Treatment of mutated neurons with rapamycin, the most widely used inhibitor of mTORC1 that reduces protein synthesis and increases autophagy, preserved ATP levels, decreased the altered AMPK phosphorylation and mitigated the effects of glutamate toxicity [89]. 

Epicatechin, a catechin derivative member of flavonoids’ family, has been associated with various beneficial effects on mitochondrial functioning [167,168,169]. In a recent report, (+)-Epicatechin was identified as a selective inhibitor of the hydrolytic activity of ATP synthase [170]. In mutated fibroblasts, blocking the ATP hydrolytic activity of Complex V with (+)-Epicatechin prevented the waste of ATP, increasing the cellular energy availability [170].

The abnormally high MMP observed in m.9185T>C NPCs has been used as readout for a high-content screening of FDA-approved drugs. In this study avanafil, a Phosphodiesterase-5 (PDE5) inhibitor, was identified as the best molecule for MMP rescue and recovery of the mitochondrial calcium homeostasis in both mutant NPCs and neurons [109].

Supplementation of L-arginine and L-citrulline, nitric oxide precursors, was proposed for the treatment and prevention of metabolic stroke in mitochondrial diseases, with clinical benefit also for pediatric patients with Leigh syndrome [171]. Indeed, low citrulline and/or elevated C5-hydroxyacylcarnitine levels were found in newborn screening of children with *MT-ATP6* variants [153,172,173]. Supplementation with citrulline and other mitochondrial cofactors, such as biotin, CoQ_10_, and the B50 complex, together with other early interventions, have been proposed to reduce the risk of decompensation and to ameliorate long-term outcomes [173]. 

### 5.2. Allotopic Expression Strategy

The proteins encoded by the mt-DNA are expressed and translated into proteins within the mitochondrial matrix. The allotopic expression of a mitochondrial gene is an approach aimed at relocating a wild-type copy of the mutated gene into the nucleus. Of course, the wild-type gene must express a polypeptide containing an amino-terminal presequence for its import into the mitochondria [174]. This strategy is intended to complement the mutant protein with the wild-type counterpart and restore the altered biochemical pathway, thus allowing treatment of the disease by gene therapy. For the success of this approach, several challenges have been assessed and, in the last three decades, handled and improved. These include the different codon dictionaries and preferences between the mitochondrial and nuclear systems, the requirement of a mitochondrial targeting sequence (MTS) for translocation within the organelle, and the need for functional integration of the polypeptide into mitochondrial Complexes [174,175].

The first successful demonstration of allotopic expression of an optimized *MT-ATP6* gene fused with the COX8 MTS in m.8993T>G cybrids was published in 2002, where both growth and ATP synthesis were improved [76]. To follow, several groups reported allotopic expression of both ATP6 and, although to a lesser extent, ATP8 proteins in human cells, trying to optimize the codon sequence for nuclear use and MTS type [40,69,176,177,178]. Indeed, the mitochondrial surface localization of nuclear-encoded mRNA allows the simultaneous translation of the polypeptide and its translocation within the organelle, a fundamental property, especially for highly hydrophobic proteins, such as ATP6. Thus, the use of the MTS and the 3′UTR of a nuclear gene such as SOD2 was tested to optimize the mitochondrial import of ATP6 in mutated fibroblasts [69,178]. In cybrids, the m.8529G>A variant resulted in the complete absence of ATP8 and a partial reduction in ATP6, leading to a defect in Complex V assembly and functionality [39]. The allotopic experiments with these homoplasmic cybrids showed that only the co-expression of the two wild-type genes, *MT-ATP6* and *MT-ATP8,* rescued the assembly and activity of CV, the OCR, and the cell viability in galactose-media [40]. Transgenic mice that allotropically express the wild-type or the m.8993T>G mutated ATP6 have been developed as a model for mitochondrial disease research, with implications for the development of DNA-based therapy [179,180].

As a promising therapy for mitochondrial diseases, further efforts have been made to improve this approach both in terms of efficient mitochondrial localization and codon sequence. Indeed, Chin and colleagues have developed an unbiased, high-content imaging-based screening platform to optimize the allotopic expression of eight mitochondrial proteins by combining thirty-one MTS and fifteen 3′UTRs [181]. Following this step, chemically modified mRNAs (modRNAs) encoding optimized ATP6 were able to restore respiration and growth under stress conditions of cybrids harboring the m.8993T>G variant, proving that modRNAs can be an alternative in the field of mitochondrial disease therapy [181]. On the other hand, Lewis et al. redesigned the sequence of the 13 protein-coding genes of the human mt-DNA by codon optimization, improving both the steady-state mRNA levels and the protein amount [182]. Indeed, only the codon-optimized subunit ATP8 protein is produced with stable expression in m.8529G>A cybrids and, when expressed in mouse model cells, was able to integrate into Complex V and rescue growth under stress conditions [182]. 

### 5.3. Nuclease-Based Approaches

An additional therapeutic option for the treatment of mitochondrial diseases aims to reduce the percentage of mutation load below the threshold for manifestations of biochemical dysfunctions. This goal is achieved by using mitochondrial-targeted nucleases that generate double-strand breaks (DSBs) at specific mt-DNA sequences [154,183]. Following DSBs, the mutant mt-DNA molecules are degraded, and the remaining wild-type molecules can replicate and repopulate the treated cells. This strategy has been validated in a number of different human cells and mouse models of mitochondrial diseases [154,183].

Mitochondrial-targeted restriction endonucleases (mitoREs) require the presence of a naturally unique restriction site that encompasses the point mutation or deletion [154]. The use of mitoRE has been tested by two different research groups in cybrids carrying the m.8993T>G variant, which generates a restriction site recognized by the enzymes SmaI and XmaI [184,185]. These mitoREs were effective at shifting mt-DNA heteroplasmy in a time- and dose-dependent manner, rescuing the intracellular ATP level and mitochondrial membrane potential [184], growth capability, OCR, and ATP synthesis rate [185]. The limitations of the mitoREs approach are that very few pathogenic variants introduce a unique restriction site into the mitochondrial genome and that the enzymes cannot be modified for a different DNA-binding specificity.

Later, great attention has been given to engineered nucleases, such as mitochondrial-targeted zinc-finger nucleases (mtZFNs) and mitochondrial-targeted TALE nucleases (mitoTALENs). In addition to the MTS, these nucleases present a sequence specific DNA-binding domain and a functionally dimeric sequence nonspecific endonuclease domain from the RE FokI that generates DSBs. The main advantages over mitoREs are that mtZFNs and mitoTALENs can be engineered to obtain different protein–DNA interactions, and thus these systems can target different mt-DNA variants. Secondly, the endonuclease becomes active only when both monomers are bound to the target DNA sequence, limiting the off-targets [154,183]. 

The proof of principle of the ZFNs approach was first provided in a work of an engineered zinc finger peptide with a methylase activity specific to m.8993T>G [186]. The system was improved to generate an initial mtZFN version that cleaved dsDNA for heteroplasmy shift in the mutated cybrids, but some unwanted mt-DNA depletion and cytotoxic effects occurred [187]. To follow, an ameliorated heterodimeric mtZFN was more effective in increasing the wild-type mt-DNA genome from 7% to 17% in the m.8993T>G cybrids, without undesired secondary effects [188]. Finally, the optimized version of this mtZFN induced an almost complete directional shift of mt-DNA heteroplasmy, with a consequent improvement in OCR, energy charge, and changes of intracellular metabolites, as shown by metabolomic analyses [189]. 

The mitoTALEN technology has been applied to reduce the m.9176T>C mt-DNA variant in artificially generated murine oocytes carrying the mitochondrial genome of patient-derived cells, preventing the transmission to offspring [190]. 

Recently, both mitoTALEN and mitoZFN have been tested in in vivo models of other mt-DNA variants [154]. All these findings may be useful in the translation of these therapeutic tools into clinical trials, although some challenges, such as immunogenicity of the delivery system and off-target effects on the nuclear and mitochondrial genome [154,183], remain to be overcome and will require further optimization.

## 6. Conclusions and Perspectives

The increasing amount of variants affecting the *MT-ATP6* and *MT-ATP8* genes is reflected in a progressively larger spectrum of clinical phenotypes associated with ATP synthase dysfunctions. Over the last two decades, the knowledge of the cellular and biochemical consequences of these variants has made great progress, thanks to the use of different cellular models, which we have reviewed. Although effective therapy for the neurodegenerative diseases related to the above variants is still lacking, a deeper understanding of the biochemical dysfunctions at both the molecular and metabolic levels has allowed us to propose interventions that could help limit the severity of these disorders. The encouraging results from pre-clinical in vitro studies are graphically summarized in Figure 6. 

Consistent and absolute findings on the consequences of *MT-ATP6* and *MT-ATP8* variants were difficult to define for several reasons, including (1) heterogeneity of tissues and cell types used in the experimental studies; (2) inconsistency of results on heteroplasmy, ATP synthesis, and respiration, particularly in intact cells; (3) difficulty in comparing genetic and biochemical analyses performed with different methods; and (4) a lack of data on the time dependence of the patient’s clinical status and related in vitro analyses. Future advances in this field will depend on identifying and standardizing experimental methods and conditions that influence biochemical and genetic assays, thus allowing potentially confounding factors to be removed. Furthermore, further work appears necessary to monitor in vivo what the ATP/ADP ratio is in the organs of patients carrying the most common variants with the aim of testing whether the data can match the experimental results in vitro. This could be achieved, for instance, with the development of high-resolution magnetic resonance spectroscopy and imaging.

However, the recent advances in the field of gene therapy, aimed at expressing the wild-type protein or reducing the mutant mt-DNA, represent a feasible option, although some challenges remain to be overcome. It should be noted that the positive results of the gene therapy trial for the treatment of Leber Hereditary Optic Neuropathy (LHON) caused by ND4 variants [162] have shown the potential of an allotopic therapeutic strategy for other mt-DNA-related diseases, including the disorders caused by *MT-ATP6*/*MT-ATP8* variants. We hope that this review contributes to the understanding of the need to coordinate in vitro and in vivo studies on mt-DNA variants to collect all critical parameters and thus better show the direction toward more appropriate therapeutic approaches.

## Figures and Tables

**Figure 1 ijms-25-02239-f001:**
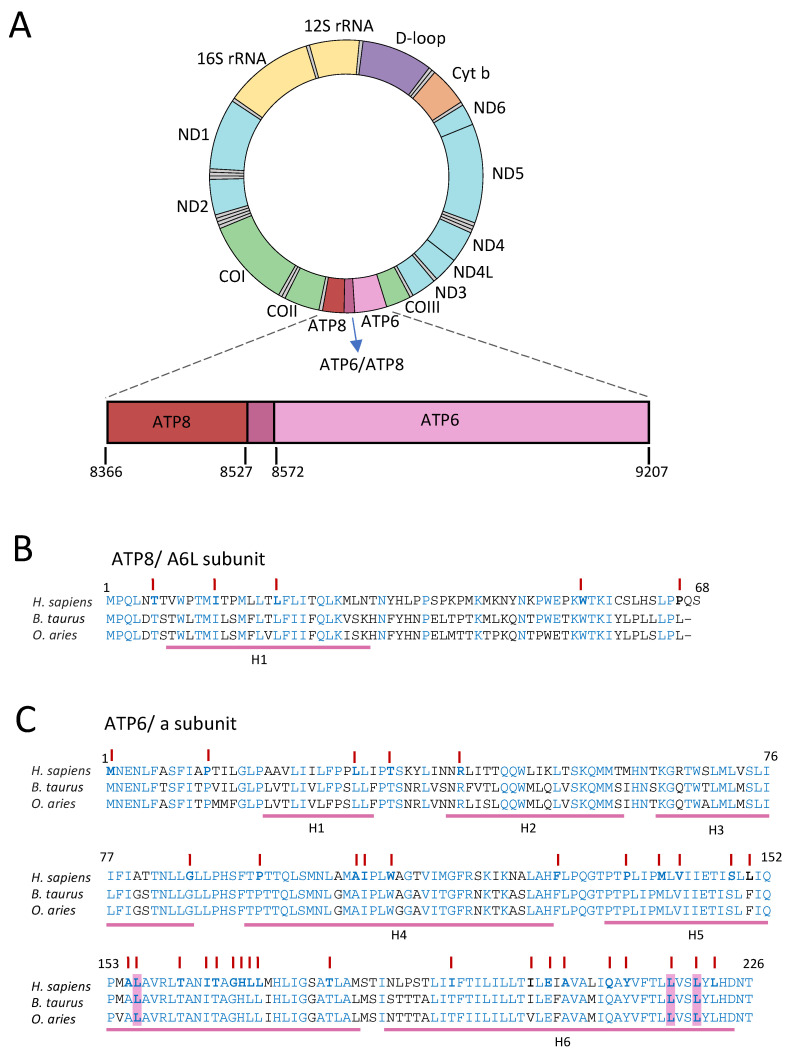
(**A**) Organization of the human mitochondrial genome. These include the non-coding control region (D-loop) (purple); 37 genes encoding 2 rRNAs (yellow); 22 tRNAs (grey); and the 13 polypeptides belonging to CI (light blue), CIII (orange), CIV (green), and CV (red and pink). An enlargement of *MT-ATP6/MT-ATP8* genes is shown, highlighting the sequence overlap (violet). Multiple alignments of ATP8 (**B**) and ATP6 (**C**) from a selection of mammalian species: *Homo sapiens*, *Bos taurus,* and *Ovis aries*. Conserved amino acids are in blue, and, at the bottom, transmembrane α-helices are indicated with pink lines according the PROMOTIF analysis performed on the human PDB structure. The three amino acid residues that are more frequently mutated in patients and are studied in detail in this review are highlighted in pink, whereas other *MT-ATP6/MT-ATP8* variants that are also shown in Table 1 are in bold and indicated by the upper red lines.

**Figure 2 ijms-25-02239-f002:**
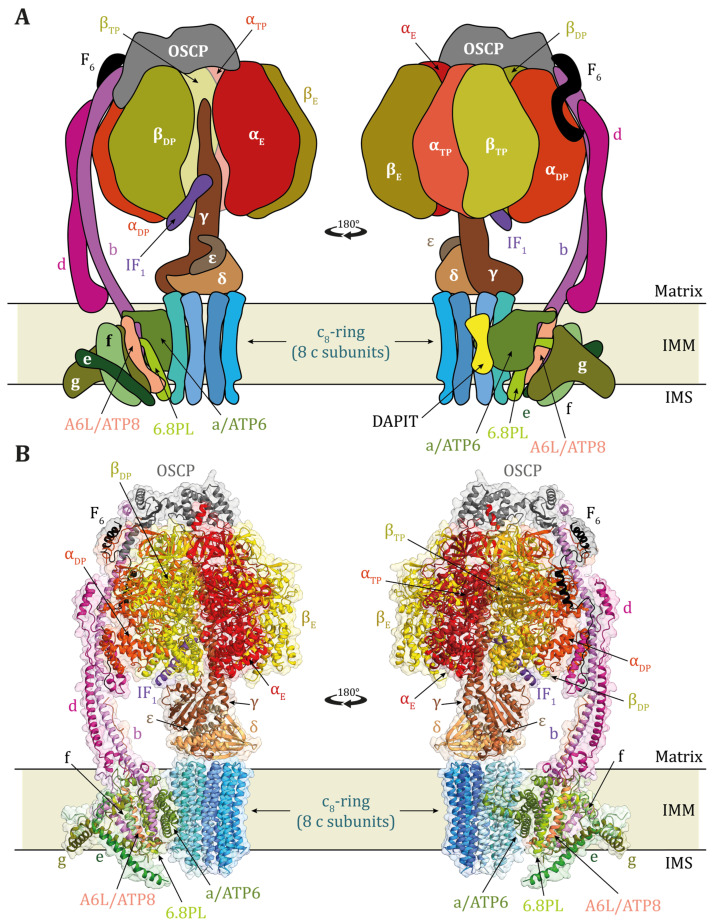
(**A**) Schematic representation of the monomeric mammalian ATP synthase, according to [16,17]. (**B**) Structure of human ATP synthase in state 1 (PDB id: 8H9S) ([16]) bound to the inhibitor protein IF_1_. The α- and β-subunits of the F_1_-catalytic domain are in different shades of red and yellow with a different labelling if the subunit is empty (αE, βE) or bound to ADP (αDP, βDP) or ATP (αTP, βTP). The γ-, δ-, and ε-subunits of the F_1_-catalytic domain are in sienna brown, sandy brown, and tan, respectively. The central stalk formed by subunits γ, δ, and ε is in contact with the c8-ring (different shades of light blue) that is part of the membrane domain and in contact with subunit a (or ATP6, olive drab). The peripheral stalk subunits OSCP, b, d, and F6 are in dim gray, violet, violet red, and black, respectively, and the A6L subunit (or ATP8) is in coral. The e, f, and g subunits in the membrane domain are forest green, pale green, and olive, respectively. The 6.8 kDa proteolipid (6.8PL) is in green yellow, and the IF_1_ inhibitor is in purple. The DAPIT subunit in yellow is not reported in (**B**) because it is not present in the cryo-EM structure in [16].

**Figure 3 ijms-25-02239-f003:**
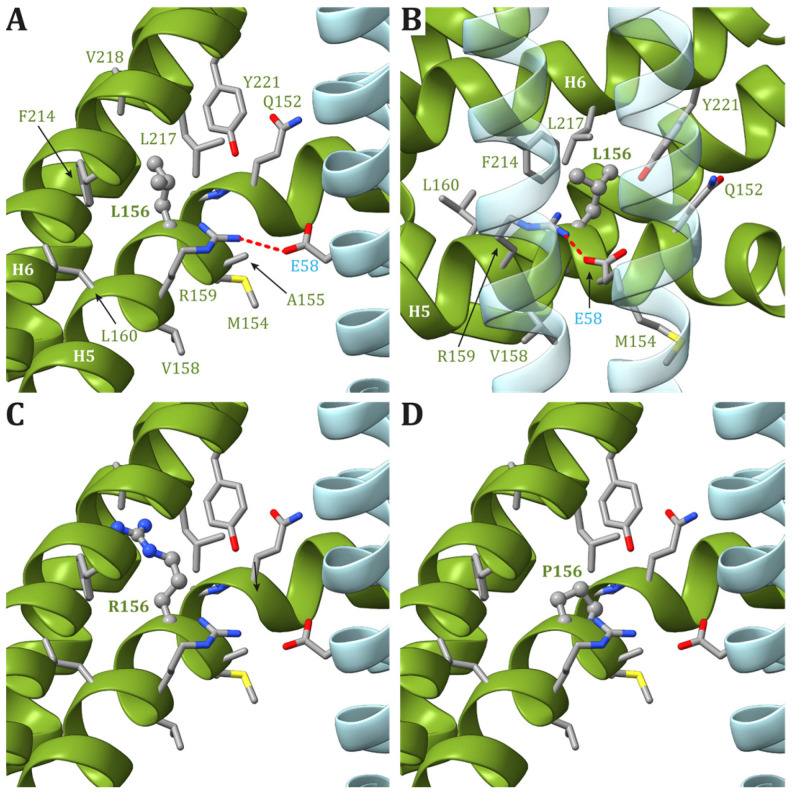
Detail of the region comprising Leu156 of ATP6 in the human structure of ATP synthase (state 1). (**A**,**B**) The native ATP6 and c subunits are reported in ribbons colored as in Figure 2. Residues labels are colored as the corresponding subunits. Leu156 is in ball-and-stick, while other residues in the vicinity of Leu156 or proposed to be part of the proton translocation process are in stick. The side chains are colored according to the atom type. The interaction between Arg159 in ATP6 and Glu58 in c_8_-ring is shown. The orientation of panel (**B**) is clockwise rotated by 90° around the vertical axis with respect to the orientation in panel (**A**). Panels (**C**,**D**) reports the model structures of the p.Leu156Arg and p.Leu156Pro variants, respectively (see Appendix A for details).

**Figure 4 ijms-25-02239-f004:**
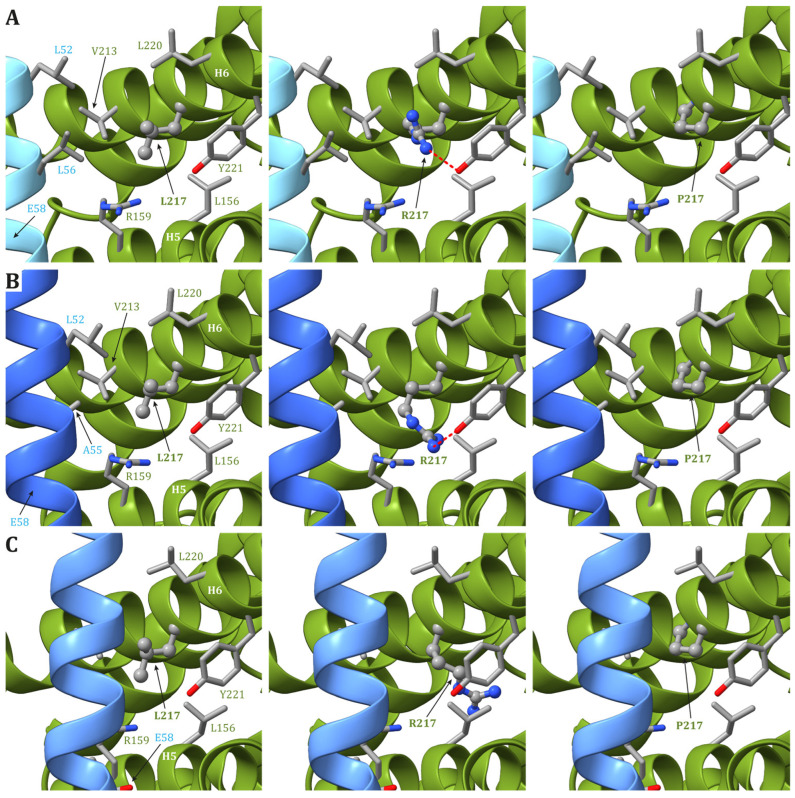
Detail of the region comprising Leu217 from ATP6 in the human structure of ATP synthase in state 1 (**A**), state 2 and 3a (**B**), and 3b (**C**). The native ATP6 and c subunits are reported in ribbons colored as in Figure 2. Residue labels are colored as the corresponding subunits. Leu217 is shown as ball-and-stick, while other residues in the vicinity of Leu217 or proposed to be part of the proton translocation process are shown as a stick. The side chains are colored according to the atom type. In the left panels, the wild-type protein is reported, while the models of Leu217Arg and Leu217Pro variants are reported in the central and right panels, respectively (see Appendix A for details). H-bonds are indicated using dashed red lines.

**Figure 5 ijms-25-02239-f005:**
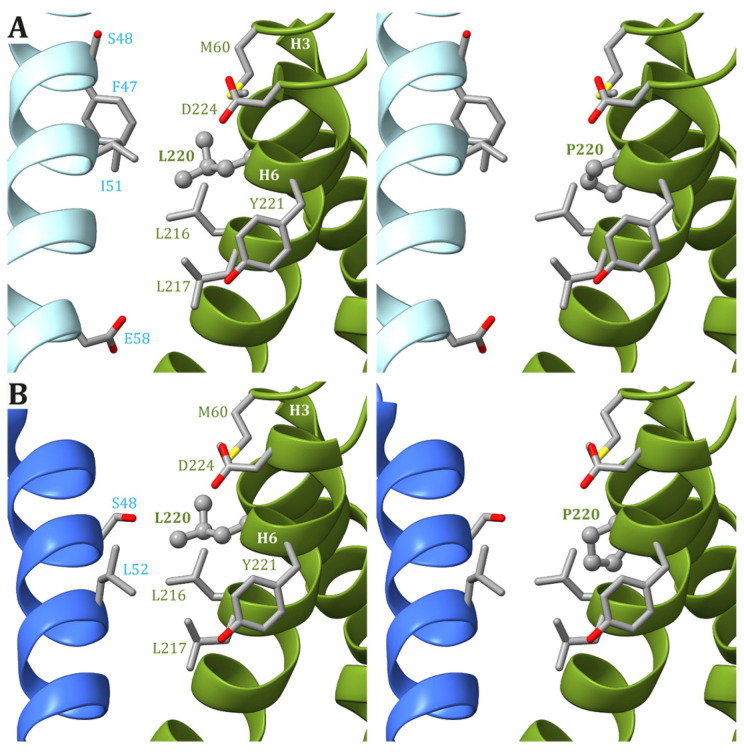
Detail of the region comprising Leu220 from ATP6 in the human structure of ATP synthase in states 1 and 3b (**A**) and states 2 and 3a (**B**). The native ATP6 and c subunits are reported in ribbons colored as in Figure 2. Residue labels are colored as the corresponding subunits. Leu220 is shown as ball-and-stick, while other residues in the vicinity of Leu220 or proposed to be part of the proton translocation process are shown as a stick. The side chains are colored according to the atom type. In the left panels, the wild-type protein is reported, while the models of Leu220Pro variants are in the right panels (see Appendix A for details).

**Figure 6 ijms-25-02239-f006:**
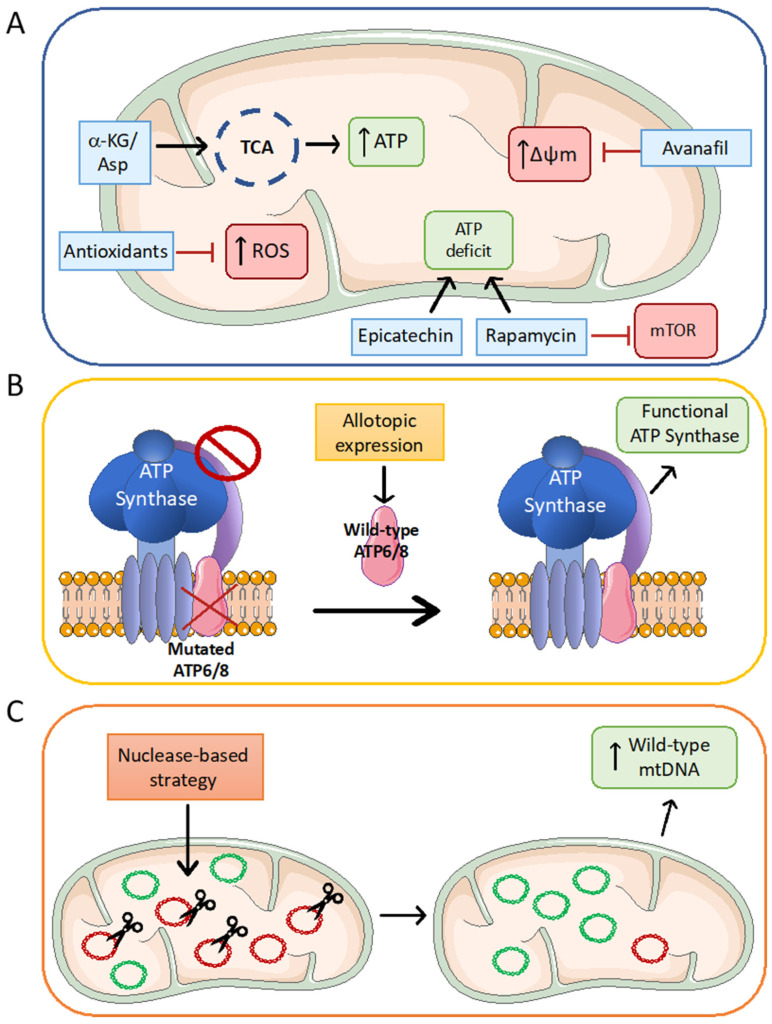
Overview of the therapeutic options proposed for the treatment of pathologies associated with *MT-ATP6*/*MT-ATP8* variants. (**A**) Drugs (blue square) targeting mitochondrial dysfunctions in mutated cells: compounds that limit an abnormally increased process (red square) or that boost a pathological reduced pathway (green square). Black arrows indicate positive regulation, while red arrows indicate negative regulation. Antioxidants or Avanafil have been reported to reduce high ROS production and mitochondrial membrane potential (Δψm), respectively. Supplementation of α-ketoglutarate and aspartate (α-KG/Asp) contributes to increasing ATP levels through reactions of the tricarboxylic acid (TCA) cycle. The limitation of ATP deficit is induced by Epicatechin or by Rapamycin. Epicatechin, by blocking the ATP hydrolytic activity of CV, increases cellular energy availability, whereas Rapamycin, by reducing the enhanced activity of mTORC1, limits the related energy-consuming processes. (**B**) Allotopic expression is aimed at replacing the mutant protein (ATP6 or ATP8) with the wild-type counterpart, restoring the ATP synthase function. (**C**) A third approach uses nucleases (black scissors) that selectively cleave the mutant mt-DNA molecules (red chains), inducing their degradation. The consequent replication of the wild-type mitochondrial genome (green chains) to maintain the copy number will increase its percentage and lead to a shift of mt-DNA heteroplasmy.

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
