# Peer review of "Variants in Human ATP Synthase Mitochondrial Genes: Biochemical Dysfunctions, Associated Diseases, and Therapies"

_ijms, 2024, doi:10.3390/ijms25042239_

Round 1

Reviewer 1 Report

Comments and Suggestions for Authors

The review from Del Dotto et al., focuses on human pathogenic variants mapping in mitochondrial DNA genes encoding ATP synthase. This review tackles a contemporary and important topic that is congruent with the concept of personalized medicine. Furthermore, this review is well-conceived and articulated in a logical and progressive manner. Therefore, this review will provide an in-depth analysis of mitochondrial ATP deficiency that will help a cadre of neuroscientists, clinicians, and physician-scientists.

However, this review needs major improvement in terms of grammar and syntax, as well as content and proper nomenclature, as detailed below. Furthermore, the authors failed to highlight the non-Mendelian transmission. This is clearly lacking. Thus, a paragraph needs to be dedicated to this with the corresponding references. Herein are included some of suggestions on how to improve syntax and grammar. Thus, this list is not intended to be complete, but rather serves as an example on how to improve the substandard English language. The whole text requires an extensive editing by a professional scientific English service provider before re-submission.

1.     Title: Please do not use the abbreviation “mt, but instead spell it out “mitochondrial”. Refer to variants instead of mutations in the title and throughout the manuscript. Amend the title so that it is grammatically correct and it accurately relates to the content of the review: “ATP synthase deficiency: Mitochondrial variants interpretation, biochemical dysfunctions, associated disorders, and therapeutical modalities”.

2.     Abstract: Mention in the first sentence that ATP synthase is also referred to as Complex V. In the first sentence, delete “human”. In the second sentence (line 12), it should read: “The mitochondrially encoded genes, mt-ATP6 and mt-ATP8, encode two subunits of the multi-subunit Complex V”. In the third sentence, t is not recommended to link pathogenic variant mapping the mt-ATP6 subunit with NARP, as depending on its heteroplasmic level it can cause NARP or MILS. Please, always refer to MT-ATP6 and MT-ATP8 throughout the manuscript with defining the abbreviation “mt” at the first usage. This comment is valid for the introduction section.

3.     Introduction: See comments above concerning the abbreviation MT-ATP6 and MT-ATP8 and refer mutations to variants.

·      Line 34: “mitochondria harbor their own circular genome (mt-DNA) containing 37 genes…

·      Line 36: define respiratory complexes; wrong punctuation: colon instead of semi-colon; capital C when referring to Complex I, etc… (correct it throughout the manuscript).

·      Line 38: “Therefore, all multi-subunit OXPHOS Complexes, except Complex II, require the nuclear and mitochondrial genomes to encode their corresponding subunits

·      Line 40: Introduce the abbreviation of reactive oxygen species (ROS). Relatively scarce presence of mitochondrial DNA repair system makes the mt genome prone to increased number of somatic variants when compared to the nuclear genome.

·      Line 43: Use the present tense

·      Line 60: State-of-the-art

·      Line 61: Clearly define what the “the first” refers to.

·      Line 68: The nomenclature of “bicistronic” is not accurate, as the mitochondrial RNAs are transcribed as long polycistronic precursor transcripts. Please amend this sentence.

·      Line 71: This review focuses on the molecular mechanisms….

·      Line 75: “Cellular experimental paradigms”

·      Table 1: The first column contains too much information, making it hard to comprehend. At the very least, ATP8 or ATP6 must be indicated instead of a number: for example, ATP8: pT6I. Normal CI-CIV activities are not mitochondrial defects, so amend the corresponding lines in Table 1. Same concern for “normal mitochondrial membrane potential. When they are many defects listed for a single pathogenic variant, it is too confusing to have them listed one after the other and separated by a semi-colon. Please, use a more friendly format, such as one defect per line.

4.     ATP synthase structure and mechanism of catalysis

·      Line 81: (F1F0-ATPase), also referred to as mitochondrial ….

·      Line 82: Delete Complex V since this has already been introduced.

·      Line 83: You need to refer to the mitochondrial membrane potential from proton electrochemical gradient in the cristae lumen. This requires a new figure to illustrate the position of the ATP synthase with respect to the other OXPHOS complexes in the cristae.

·      Line 88: spell out 8

·      Line 92: added to 10 more distinct subunits

·      Line 94: Grammatically incorrect.

5.     Biochemical Dysfunctions Related to Pathogenic Variants Mapping in the mt-ATP6 and mt-ATP8 Genes

·      Line 142: Refer to patient-derived cells instead of patient’s specimens.

·      Line 143: other cellular paradigms have led to in-depth investigations

·      Line 144: More specifically to the most common ones listed in Table 1.

·      Line 145: The syntax of this sentence is not correct, please amend it. Also, define the concept of heteroplasmy, which is introduced for the first time in the review.

·      Line 151: Use the proper nomenclature when describing the mutated AA in the protein mt-ATP6 or mt-ATP8. Also, please address the grammatical mistakes in this sentence, which is by the way too long.

·      Line 159: See comment for line 142.

·      Line 160: Are you referring to dermal fibroblasts or other types? Please clarify.

·      Line 162: Define the abbreviation OCR even though it was introduced in the legend of Table 1.

·      Lines 164-165: Fix the syntax of this sentence

·      Line 169: Incorrect syntax

·      Line 173: “Of notice” is inappropriate in this context. Please, rephrase; It is worth noting that no significant effects….”

·      Line 176: Patient-derived cells. Why are you limiting to the two listed stress conditions. Please justify these.

·      Line 178: Expand on the concept of heterogeneous mitochondrial membrane potential.

·      Line 182: Cite the appropriate references for the influence of the nuclear background on the penetrance of the pathogenic heteroplasmic variants.

·      Line 185: The authors showcase the use of cybrids as an advantage since they share the same nuclear background, that of an immortalized cell line. The main advantage of cybrids is to determine whether a new mitochondrial variant is pathogenic or not. Besides that, cybrids are not adequate to investigate the impact of a specific mitochondrial variant onto mitochondrial dysfunctions. Please amend this section to highlight the inherent limitations of the cybrid models.

·      Line 197: Define ECAR. How is an increased ECAR relevant to the biochemical phenotype? Please clarify this ambiguity.

·      Line 211: Discuss the drawbacks and limitations of iPSCs for maternally inherited mitochondrial respiratory disorders. Comment on the stability of the heteroplasmic levels of these mt pathogenic variants specific to the review. This paragraph is written in general terms instead of being pertinent to the pathogenic variants for mt-ATP6 and mt-ATP8. This section needs to be revised.

·      Line 262: “For the m.9176T>G” is incorrect. It should read “In the case of the m.9176T>G variant, a decreased ATP synthesis and mitochondrial respiration due to a defective OXPHOS pathway led to a partially disassembled Complex V accompanied by an increased in mitochondrial membrane potential in fibroblasts”. Please mention the origin of these fibroblasts: patient-derived? If yes, please provide further information on the patients

·      Line 266: Precise what Atp6p is.

·      Line 271: This sentence is too long with a poor syntax.

·      Line 376: Define the acronym MILS. Mention that MILS is lethal whereas NARP is not.

6.     Therapeutic Approaches

·      Line 417: Syntax issue.  In fact, the whole paragraph needs to be rewritten

·      Legend of Figure 6: Needs to define the following abbreviations: alpha-KG, Asp, TCA, the symbol for mitochondrial membrane potential. Also, define ATP shortage: due to what? What are the targets of the drugs epicatechin and rapamycin?

·      Lines 435-437: Grammar issue: both past tense and tense used in this sentence. Which substrate are the authors referring to?

·      Line 445: Which pathological conditions the authors are referring to? Please precise. This is critical in the context of the well-established high heterogeneous clinical phenotypes and genetic variability. The authors need to be more specific on the origin of ROS? Which OXPHOS Complexes are responsible? Again, this statement is articulated in the general context and clinical paradigms. How relevant is it to the maternally inherited mitochondrial disorders caused by the pathogenic variants targeting the mt-ATP6 and/or mt-ATP8 subunits of Complex V?

·      Lines 449-452: Do the authors refer to increased mitochondrial rate ATP synthesis or increased total cellular ATP? This needs to be expanded.

·      Line 453: Please rephrase “in NARP fibroblast”. What do the authors mean by “no biochemical studies have been performed”? Please expand. Otherwise, there is no need to mention this incomplete study. And superoxide anion should be plural.

·      Lines 454-458: This statement cannot be understood as is given the absence of information on the metabolic correlations among glutathione peroxidase, thioredoxin reductase activity and NRF1 protein. Please expand and add a figure on these complex metabolic pathways.

·      The authors need to state the limitations of these antioxidants given that most of them have not been tested in patient-derived cellular paradigms or tested in a clinical setting. This should be clearly stated at the outset of this section.

·      Line 461: Define the low dose of resveratrol.

·      Please clarify throughout this section which of these antioxidants are available OTC/prescribed versus only in clinical trials.

·      Line 463: The authors need to state the impact of EPI-743 on the pre-clinical paradigms before diving into the current clinical trials. Also, the authors need to introduce what EPI-743 is, its properties, and PK/PD characteristics. Pleas clarify if the authors refer to a pharmacological grade of EPI-473 authorized by the U.S. FDA and/or the European Medicines Agency. Please mention the ongoing phase 3 clinical trial to test vatiquinone, previously known as EPI-743, on patients with specific primary mitochondrial disorders.

·      Line 469: Please correct the grammar mistake and define which mitochondrial diseases the authors are referring to.

·      Line 470: Please introduce mTORC1 and its role in mitochondrial bioenergetic pathway. What is the origin of the “mutated neurons”? What are those?

·      Lines 476-478: Is this correct that the NCT03747328 clinical trial is ongoing? Or is this study currently on hold? The authors need to make sure of the accuracy of this information and if the status has changed, please justify why in the text.

·      Line 484: “Redout” should be spelled “Readout”. Define how the m.9185T>C NPCs have been generated and define NPCs.

·      Line 487: which disease the authors are referring to?

·      Lines 488-495: This whole paragraph needs to be rewritten as the information is not quite accurate. For example, the authors refer to Leigh syndrome, when in fact near-homoplasmic pathogenic variants mapping in the mitochondrial genes encoding the ATP6 subunit of Complex V result in MILS and not Leigh syndrome. Are the authors referring to oral administration of arginine or IV infusion of arginine? This is critical in the context of their statement on treatment and prevention of stroke-like episodes. Again, the authors remain vague in terms of which mitochondrial diseases they are referring to. Please clarify. What do you define as decompensation?

·      Line 497: Please refer to mitochondrial genome instead of mtDNA. And this is valid throughout the whole manuscript.

·      Line 498: Mitochondrial matrix instead of mitochondria.

·      Line 499: This is confusing: relocate a wild-type copy of the mutated gene into the nuclear genome. Please clarify and rephrase the principle of allotopic expression.

·      Line 501: Is it replacing or complementing?

·      Lines 540-547: Please comment on the pros and cons of the nuclease-based approach to reduce the heteroplasmic load of pathogenic mitochondrial variants accompanied by the corresponding references.

7.     Conclusions:

·      Please wrap up the conclusive remarks with future perspective.

Comments on the Quality of English Language

The English language of this paper is substandard as it contains numerous problems with punctuation, grammar, tense, and/or sentence structure and clarity. Many sentences are written with an inappropriate colloquial/jargon style and often lack logical structure. The whole text requires an extensive editing by a professional scientific English service provider before re-submission.

Reviewer 2 Report

Comments and Suggestions for Authors

In the review entitled, “Mutations in human ATP synthase mtDNA genes: biochemical dysfunctions, associated diseases, and therapies”, the authors report the pathogenic mutations in mitochondrial ATP synthase genes and highlight the molecular mechanisms underlying ATP synthase deficiency that promote biochemical dysfunctions. Overall, the manuscript needs much improvement and structuring. There are some comments which should be considered while revising the manuscript.

Comments:

1. Please highlight the importance and need for the review, particularly in the abstract and introduction sections.

2. The authors mention that mutations in two subunits i.e. ATP6 and ATP8 have been described in the review. Please provide justification for selecting only these subunits. Also mention their importance for mtDNA. Previously, Roza Kucharczyk in 2018 also highlighted the mutational role in ATP6 and ATP8; please mention the importance of the current review and how it is advantageous over/different than Roza et.al’s publication.

3. The authors also mention that the review is concerns ATP synthesis deficiency, which mainly manifests in childhood. Please define what age range was considered for selecting studies/conditions with children. Please also ensure that all the studies cited in the review are in the said defined age in children.

4. Additionally, the authors also mention that mutations in mtDNA increase with aging. Please provide details of which kind of mutations are considered in the review; genetic/inborn mutations in children or aging related mutations. Please clearly define the parameters of the review in the abstract and introduction sections.

5. In several places on page 7, there appear strange symbols. For example, line#88-89, phosphorylating sector F1 (=), and three composing Fo. Please check if this is an error and please fix the issue accordingly.

6. Paragraph 2. ATP Synthase Structure and Mechanism of Catalysis: This section is a single, very long paragraph. Please break into smaller sections/paragraphs to make it easier to comprehend.

7. More references for all the statements are needed throughout the review.

8. Only three sections on the mutations have been discussed. Each section can be broken down into clear sub sections, to increase readability. Please also explain the biological relevance of each mutation in detail.

9. The review has a lot of good information, but it needs to be structured in a much better fashion. In the current state it is very difficult to comprehend. Since the authors describe diseases associated with the mutations, it would be beneficial to sub categorize based on disease types or organ/organ systems affected. For examples, neurological, cardiac etc.

10. The therapeutics section is very difficult to follow. Please mention sub-heading to differentiate between in vitro and in vivo studies as well as human studies/clinical trials. This section can also be divided into disease types/systems affected to increase readability for the audience.

Comments on the Quality of English Language

Minor editing required.

Reviewer 3 Report

Comments and Suggestions for Authors

The article is an interesting overview of mitochondrial dysfunction caused by mutations in the genes encoding ATP-synthase. The authors focused on a specific region of mitochondrial DNA and covered the problem well. After corrections, the article can be published.

The comments include requirements and recommendations.

1) ‘The three mutated amino acid…’ (line 54): What do you mean when you use this term? It seems that you did not choose a proper term. I can guess what you mean but still, you have to rephrase the sentence.

2) Table 1 (line 57)

You have not deciphered the following abbreviations: ‘CV’, ‘CI’, ‘CIV’, ‘CIII’.

!!! Order the reference numbers according to their occurrence throughout the article. After the references [4,6–8] in the first paragraph of the Introduction (line 45) the reference [31] shows up in the Table 1. It looks like Table 1 was moved from another part of the text since it contains references from [31] to [109] following the references [9,10] (line 67) within the text. Please synchronize them.  

‘Normal [31]12/29/2023 9:46:00 AM’: Seems like it's redundant.

You should better explain how the percentage of heteroplasmy is calculated.

3) Having problems reading the symbols in the following lines: 89, 100, 107, 111, 114, 116.

4) ‘Recently, an additional binding site of IF1 on the ATP synthase has been described [21]’ (lines 97-98): Uninformative sentence, perhaps worth adding more details.

5) Figure 6 (line 424): Poor image resolution. The figure should be read separately from the text, add a description of the abbreviations used in Figure 6.

Round 2

Reviewer 1 Report

Comments and Suggestions for Authors

1.  This is grammatically incorrect to use arabic numeral in a sentence when numbers are inferior or equal to 10, unless there is a number above 10 mentioned in the sentence. Please correct this grammar error throughout the manuscript.

2. The authors need to apply the 2020 updated terminology in Genetic that was approved by the prominent professional organization (American College of Genetics and Genomics). The new terminology the term "mutation" has been replaced by "variant" in the medical genetics' literature due to the negative connotation of mutation as well as the fact that most variants do not represent new mutations. More specifically, variant classifications are applied to single gene conditions and include the following categories: pathogenic, likely pathogenic, uncertain significance, likely benign, or benign.

3. Former line 487:  which disease the authors are referring to?
Author’s reply: Again, all the mitochondrial diseases are listed in the review cited, which is available for details. I highly recommend the authors to briefly precise which disease the authors are referring to without the readers to go read the corresponding reference. Thus, the revised version must include some details to facilitate the reader's overall comprehension and alleviate any ambiguity.

4. MILS is a subset of Leigh syndrome. However, MILS and Leigh syndrome are clinically different for many reasons, one of them being the time of onset. Thus, MILS and Leigh syndrome are considered clinically different when Leigh syndrome has an adult onset. It goes well beyond the genetic origin of the pathogenic variant. The authors need to clarify their statement.

Comments on the Quality of English Language

Minor editing

Round 3

Reviewer 1 Report

Comments and Suggestions for Authors

The authors adequately addressed the remaining concerns encapsulated in the previous critique.